# Web Page Content Block Identification with Extended Block Properties

Kiril Griazev * and Simona Ramanauskaitė

Department of Information Technologies, Vilnius Gediminas Technical University, Sauletekio al. 11, LT-10223 Vilnius, Lithuania
* Correspondence: kiril.griazev@vilniustech.lt

**Abstract:** Web page segmentation is one of the most influential factors for the automated integration of web page content with other systems. Existing solutions are focused on segmentation but do not provide a more detailed description of the segment including its range (minimum and maximum HTML code bounds, covering the segment content) and variants (the same segments with different content). Therefore the paper proposes a novel solution designed to find all web page content blocks and detail them for further usage. It applies text similarity and document object model (DOM) tree analysis methods to indicate the maximum and minimum ranges of each identified HTML block. In addition, it indicates its relation to other blocks, including hierarchical as well as sibling blocks. The evaluation of the method reveals its ability to identify more content blocks in comparison to human labeling (in manual labeling only 24% of blocks were labeled). By using the proposed method, manual labeling effort could be reduced by at least 70%. Better performance was observed in comparison to other analyzed web page segmentation methods, and better recall was achieved due to focus on processing every block present on a page, and providing a more detailed web page division into content block data by presenting block boundary range and block variation data.

**Keywords:** web segmentation; hierarchical segments; web page labeling





## 1. Introduction

The vast majority of data are presented in web systems in HTML format. The purpose of designing this technology was to present data to humans. However, current technologies are increasingly interconnected, so the content must be designed for machines to read, rather than just humans [1]. Machine-adapted labeling of content on web pages is a base for automated content extraction, data mining, content transformation, and other needs [2]. However, the existing HTML standard is slowly moving away from presentation over data. Content-related semantic tags like header, nav, section, article, figure, etc., are introduced but are mixed with general-purpose tags to build the needed design. Thus, web page content block identification is a relevant task that should be automated rather than performed manually.

For automated data gathering from web pages, after the HTML code is obtained, it should be divided into content blocks and then the type of each block should be defined [3]. The first part is conducted by segmenting the HTML code or dividing it into content blocks. The second part uses intelligent solutions to classify the blocks into predefined types [4]. However, both of these parts currently are facing some challenges, which are related to the hierarchical nature of web page blocks [5]:

- Some content blocks might be divided into smaller internal blocks. For example, menu can have menu items, and article paragraph can have some highlighted words of phrases. In the first case, the division into internal blocks is meaningful, while in the case of text formatting, it mostly will be redundant. Therefore, the level of detailing might cause redundancy of blocks.

- Content block might have HTML code ranges, caused by its presentation structure. If the block content is surrounded by several tags, it can be gathered by using different selectors, and the path to the segment and its content might vary. Therefore, for more accurate block classification, knowledge of the possible ranges would be beneficial.
- Hierarchical and sibling relations between different blocks might positively affect the block classification accuracy. Blocks like navigation menus and lists have a hierarchical structure, therefore keeping the links between the blocks would bring more features for correct block type identification. At the same time, identification of relations between blocks might lead to a reduced size of the dataset without losing any of the data. Only one sibling element could have a label, while the remaining ones could be associated as sibling blocks of the same type.

The mentioned issues in conjunction with other limitations affect the fact there is no solution capable of automatic extraction of unknown structure website data and linking it to the appropriate content type [6]. The existing solutions require a predefined website structure or extraction of specific data, such as listed items, links or other blocks only.

Our goal in this paper is to propose a novel approach to web page block identification, which would provide a bigger variety of content blocks and more detailed information about the content blocks, in comparison to existing web page segmentation solutions. Each identified block should contain a content reflecting certain structural element of the web page (menu, header, title, contacts, paragraph, etc.). The method should identify all content blocks with its internal structure and provide more detailed block data than traditional web page segmentation solutions, such as block's ranges, hierarchical relations, and siblings. Such an extension of content block information would extend the capabilities of the segmentation data application and lead to a better website content block classification.

## 2. Related Work

Currently, the content block extraction from web pages is mostly conducted by web page segmentation methods [7,8] as the main part of content blocks match the web page segments, while selected segments can be repeatedly analyzed to get internal structure of segments of it. A significant portion of existing solutions for web page segmentation is based on visual page segmentation. M. Cormier et al. [9] and J. Zaleny et al. [10] rely on visual analysis only, eliminating the dependency on web page implementation technologies. However, image segmentation-based solutions usually are more expensive in computational time in comparison to document object model (DOM)-based methods [11]. J. Kiesel et al.'s [12] research indicates that segmenting web pages visually provides high performance. However, in the research, the Vision-based Page Segmentation (VIPS) method, which uses both DOM and visual segmentation solutions, had the highest performance.

By integrating DOM tree analysis, a wide range of metadata can be analyzed. We can, for example, apply text analysis to identify related text on a web page [13], detect malicious websites [14], and segment blocks [15]. In addition to text, it also includes content analysis and text density for web page segmentation [16].

Furthermore, DOM structure and its related features are also relevant for web page segmentation [17,18]. Language-independent solutions for dedicated content extraction are available [19]. Some existing web page segmentation solutions are oriented to specific application areas. For example, A. Sonaja and S. Gancarski [20] proposed a solution to convert HTML code from version 4 to version 5. The project involves web page segmentation and its migration to HTML 5. Image segmentation is used to identify different types of images on a web page [21]. However, the existing solutions do not provide a means of obtaining complete data on the web page block, which would be necessary for extending web mining capabilities [22]. Existing methods do not focus on the range of HTML code corresponding to the same block (additional tags can be used to surround the content and affect the presentation and variety of selectors or paths to extract the content). Block variations (relationships between siblings or hierarchically related blocks) are not linked to obtaining a more interconnected block map as well.

To compare the existing segmentation methods, several datasets are prepared. One of the most used was created in 2014 and presents a list of web pages that were popular at that moment [23]. The labeling of the dataset presents a few main blocks on each web page with no details on composite elements. On average, this dataset has 13 labeled blocks for each web page, while the median is 16 blocks. The summary of accuracy metrics using different segmentation methods is presented in Table 1.

**Table 1.** Summary of proposed method accuracy metrics and comparison to other methods.

| Method | Precision | Recall | Accuracy | F-Score |
|---|---|---|---|---|
| BoM [16] | 31% | 26% | 26% | 28% |
| VIPS [17] | 24% | 26% | 24% | 25% |
| SegBlock [18] | 38% | 40% | 38% | 39% |
| Semantic-Block [19] | 40% | 43% | 42% | 42% |
| Fusion-Block [20] | 45% | 54% | 48% | 49% |
| Integrated-Block [20] | 52% | 62% | 54% | 53% |

Results of other methods were gathered from previous research papers [22] and include the following web page segmentation methods:

- BoM [24] combines the structural, visual, and logical features of web pages.
- VIPS [25] is visual analysis of web pages only.
- SegBlock [26] combines the visual appeal, logic, and features of the content on a web page.
- Semantic-Block [27] uses Gestalt laws.
- Fusion-Block [28] is Gestalt law-inspired and subsequential re-segmentation, which uses semantic text similarity.
- Integrated-Block [28] uses DOM structure, is vision-based, and uses text-based similarity metrics analysis based on web page segmentation.

Research works on specifically web page content block identification exist as well [29,30]. Those are able to identify main web page structure blocks with almost perfect accuracy, however, they are oriented on content block identification based on an analysis of multiple web pages in the same website. This approach is not suitable for one page websites or those, who use different design for different sections or even pages. As well the block bounds or block variations are not estimated in those early web page content block identification solutions. Meanwhile the relations of similar DOM elements is an important aspect [31]

Another web page content block identification direction in research papers is search for some specific content block in the web page [32]. In most of the cases it is based on the content block text analysis. However the full potential of the area is limited because of lack of high quality datasets, suitable for machine learning based models.

The existing datasets and methods are mostly oriented on web page segments, which usually identify segments, not content block. One segment can combine multiple content blocks into bigger, visually consistent segment. Despite the fact, the dataset has just segments, main content blocks, and no details of internal blocks, or code ranges of the blocks, the accuracy metrics are far from perfect. This illustrates the area is complex and requires different solutions to fully master the web page content segmentation and division into content blocks.

## 3. Block Identification Method for Block Range and Variation Estimation

### 3.1. Definition of Block Range and Variation

HTML code has a hierarchical structure, where one tag contains other tags that define smaller components of the parent tag. An example of a hierarchical relationship can be seen in Figure 1. Menu elements are blocks of information that can be visually identified on the website. They also have smaller components—menu items. Menu items are arranged according to the hierarchy of the menu element. Additionally, those menu items are siblings since they are presented in the same block and have the same structure.

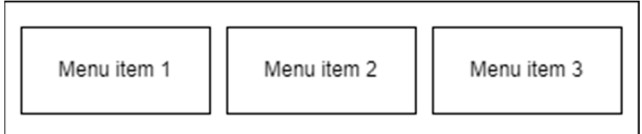

**Figure 1.** Example of block (menu) with three hierarchically related inner blocks (menu items).

With different tags and their combinations, the example menu case with multiple menu items can be realized. Figure 2 shows one of the examples. It demonstrates how the boundaries of the menu can be indicated by different tags (<header>, <div> or <nav>). All three possible boundaries visually produce the same block. In all cases, the user would treat it as a menu regardless of the possible boundaries of the menu. However, in some cases (machine learning, block structure matching, etc.) the usage of specific or all available tags can affect the desired result. To extend machine-oriented block segmentation properties, we define the range (maximum and minimum boundaries) of HTML blocks.

```html
<header><!-- maximum boundary start for block B1 -->
    <div class="container">
        <section><!-- minimum boundary start for block B1 -->
            <div><!-- maximum/minimum boundary start for block B1.1 -->

            </div><!-- maximum/minimum boundary end for block B1.1 -->
            <nav><!-- maximum boundary start for block B1.2 -->
                <ul><!-- minimum boundary start for block B1.2 -->
                    <li><a href="page1.html">Menu item 1</a></li><!-- B1.2.1 -->
                    <li><a href="page2.html">Menu item 2</a></li><!-- B1.2.2 -->
                    <li><a href="page3.html">Menu item 3</a></li><!-- B1.2.3 -->
                </ul><!-- minimum boundary end for block B1.2 -->
            <nav><!-- maximum boundary end for block B1.2 -->
        </section><!-- minimum boundary end for block B1 -->
    </div>
</header><!-- maximum boundary end for block B1 -->
```

**Figure 2.** HTML code example, illustrating the range between the maximum and minimum boundaries of the blocks.

The maximum boundary is the tag (element in the DOM tree), which defines the widest possible area, covering only the content of the block. In the DOM tree, it would be the highest element, containing only the content of the block and its child blocks, excluding siblings. Meanwhile, the minimum boundary is the tag, which will produce a child block with just partial content of the block, not the full content. If it were in the DOM tree, it would be the last element, before the child elements are visible.

In the example, presented in Figure 2, <section> tag contains both image and the navigation items, therefore it is a minimal bound of block B1. Consequently block B1.1 has matching maximum and minimum boundaries as going deeper than the div would produce the content itself (image) and going wider would include other sibling block content. The area between minimum and maximum boundaries of each block is the part, which has no content in it.

Block variations are defined as similar blocks belonging to the same parent block. In the case of menu items, as shown in Figure 2, the three menu items are variants of each other. Not all child tags are variants of one another (blocks B1.1 and B1.2 are not variants, just siblings as has completely different structure). Whether those siblings are variants or not depends on their structural similarity. The variant blocks must share the same parent block and internal structure, but not the content. In case of Figure 2 example, the path of tags for all B1.2 child elements is the same (<header>,<div>,<section>,<nav>,<ul>,<li>,<a>), while the path between B1.1 and B1.2 is different. The level of similarity can be adjusted based on the web page type or segmentation requirements.

*3.2. General Idea of Web Page Division to Content Blocks with Extended Properties*

Combining the above two features (block boundaries and variations) we can achieve additional flexibility in the identified content block data in both manual and automated



labeling processes. This additional flexibility comes from the fact that by using these features we can reduce the total number of blocks per page by using relationship connections between them. That way when a user is manually marking blocks he or she only needs to label one variant of similar blocks, while the others will be picked up automatically. This would require less work than labeling the whole page.

Figure 3 shows an example of how block relationships make a difference. If we had no block relationships then this code snippet alone would have 8 distinct blocks that would need to be identified to fully process this code snippet (numbered in red circles). To extract only the content-filled blocks, without taking the boundaries into account, we would need to label fewer blocks; four would be sufficient—one for the menu (red dot no. 1) and three for its items (red dot no. 4, 6 and 8). But this can lead to results that are more difficult to verify even if they are correct. The difficulty arises from the fact that during validation only exactly labeled blocks would be deemed correct, so that approach requires a lot of precision. Through the use of block relationships, we can greatly reduce the number of distinct blocks. However, we are also able to maintain information about the complete structure that allows the extraction of all data. With relationships, we would technically have only 2 blocks (marked in blue circles—one for menu and one for menu item) while all other related blocks would be accessible either by hierarchical relation or by structure variations.

```
 1  <nav>  1
 2      <ul>
 3          <li>  2
 4              <a href="#link1">Link 1</a>
 5          </li>
 6          <li>
 7              <a href="#link2">Link 2</a>
 8          </li>
 9          <li>
10              <a href="#link3">Link 3</a>
11          </li>
12      </ul>
13  </nav>
```

**Figure 3.** HTML code example, illustrating the reduction of labeled blocks when instead of 8 blocks (marked with red dots), a person needs to assign labels to two (marked with blue dots) of them.

To implement web page division into blocks with extended properties, we analyze the DOM tree from top to bottom, starting with the <body> tag (see Algorithm 1). This tag would represent the most general block—web page content. The estimation of block boundaries will provide additional value, as the minimal boundary will define where the actual content and inner blocks start. At the same time, the relationship between child blocks will be able to estimate the repeating structures of variant blocks.

The web page division into blocks solution was implemented to match the dataset data structure provided in our earlier research [22]. It takes an URL address as input and stores all the identified blocks, and the relations between them, in the database. The methods responsible for block maximum and minimum boundary identification and variation estimation are presented in further sections.

---

**Algorithm 1**: segmentation

---

    **input**: DOM tree of the rendered web page HTML code
    **set** analyzed block to <body>
    **set** analyzed parentBlock to <body>
    **if** analyzed block is not empty **then**
        **call** boundaryEstimation with block and parentBlock **return** minBlock
        **call** getChildren with minBlock **return** children
        **call** getVariations with children
        **for** each children
          **call** segmentation with children
        **end**
    **end**

---

*3.3. Method for Block Range Estimation*

Due to various design requirements blocks that store the same information, but have different visual representations can often use different HTML structures. Additional HTML elements may be required solely to achieve the required visual representation. As a result, it is crucial to detect the minimum and maximum boundaries of a content block.

As the HTML is analyzed from top to bottom, we can assume the block starts with the maximum block boundary. Deeper elements are analyzed to find the bare minimum. To achieve this we calculate content similarity while traversing the HTML tree. The maximum boundary of Figure 2 starts at line 1 and ends in the last line. The whole code represents the maximum boundary.

The minimum boundary of this block is defined by the <section> element. Comparing the content of these 2 boundaries, we would get the same content that the user sees in both cases. Only the actual data that the user would see, ignoring any other elements, carrying no information, is used for block content similarity estimation.

There are times when small blocks of content can be added to create a visual effect without impacting the content. The symbols can be applied to separate elements (at the beginning or the end of a menu, etc.). A similar problem is addressed by F. Fauzi et al. [33] only meaningful images are extracted, ignoring non-relevant images. We used the Hamming normalized distance [34] to measure the similarity between the content of the blocks to account for noise in the content. The content was extracted by stripping HTML blocks and leaving only clean text for comparison. The threshold value was set at 0.1, by analyzing existing tendencies in web pages. Any comparison of parent and child tags that produces a value below 0.1 means that we still haven't found the minimum boundary. As soon as we get a comparison value of 0.1 or above we know that the minimum boundary was reached during the previous iteration. All blocks between the maximum and minimum boundaries (including the boundary blocks) are saved in the dataset as blocks' length boundaries.

One of the disadvantages of this approach is that it does not cover blocks that have no text content. Such situations can occur when self-closing HTML tags are used. One of the most common self-closing tags is  (see Figure 4). Its data are all stored in attributes, thus the content of such a block is empty after HTML tags are removed from the text. Another situation where this issue can arise is when content is added via CSS rules. To represent links to the corresponding website, social media icons (logos of social media networks) are commonly used. In such cases, it is often an icon being applied to an HTML tag via CSS rather than via HTML. This would again result in no content in the HTML tag. But these are edge cases that deal mostly with visual information.

```html
1  <section><!-- maximum boundary start -->
2      <div>
3          <div><!-- minimum boundary start -->
4              <div><!-- minimum boundary start if image is ignored -->
5                  <h1>Heading</h1>
6                  <p>Paragraph</p>
7                  <a href="url"><i class="fa fa-play"></i> Link text</a>
8              </div><!-- minimum boundary end if image is ignored -->
9
10                 
11         </div><!-- minimum boundary end -->
12     </div>
13 </section><!-- maximum boundary end -->
```

**Figure 4.** HTML code example, illustrating the range between maximum and minimum boundaries when empty content elements are included.

To capture such blocks correctly, we added additional checks to the algorithm. First, we check whether the parent block contains any content. If there's no content and the

analyzed block is the only child block, then we can safely assume that the child block can be added as the block length boundary.

When analyzing blocks that have content we should still check for images to make sure that the correct block boundaries are determined. To do this we count the occurrences of image tags within parent and child blocks. This is only conducted when the textual content of parent and child blocks is the same and there is more than one child block present. In the simplified HTML code example in Figure 3, we present a case when an incorrect block length variation can be captured. This is because image-like blocks are not accounted for. If the image tag is ignored in this example, then the minimum content block boundary would be incorrectly determined and increased by 1 level, compared to the correct detection.

The schema of HTM block boundary range estimation is presented in Algorithm 2. It takes into account content similarity and the existence of  tags. For each candidate to the minimal boundary, the method will be called recursively, and for each child block accordingly, while traversing all DOM trees from top to bottom.

---

**Algorithm 2**: boundaryEstimation

---

    **input**: block for analysis and its parent block
    **set** minBoundary to block
    **set** maxBoundary to block
    **set** distance to 0
    **repeat**
      **call** getChildren with block **return** children
      **if** parentBlock clean text is empty **then**
        **if** number of children <= 1 **then**
          **set** distance to 0
        **else**
          **set** distance to 1
        **end**
      **else**
        **if** parentBlock clean text = block clean text and number of children >1 **then**
          **call** getImageCount with parentBlock **return** parentImages
          **call** getImageCount with block **return** blockImages
          **if** parentImages = blockImages **then**
            **set** distance to 0
          **else**
            **set** distance to 1
          **end**
        **else**
          **call** hammingDist with block text and parentBlock text **return** distance
        **end**
      **end**
      **set** block to first children
      **set** minBoundary to first children
    **until** distance >= 0.1
    **store** block data with minBoundaries and maxBoundaries
    **store** block data with children relations
    **retrun** minBoundary

---

### 3.4. Method for Block Variation Estimation

Block type variation means that we're identifying blocks of the same purpose but with different content. This would allow us to identify clusters of blocks of the same type. Clusters in this case are defined by a common parent block. To achieve this we traverse the HTML tree and look for adjacent HTML blocks that have a similar HTML structure. Structure similarity is evaluated with the help of the HTML path similarity estimation algorithm. When traversing the HTML tree, we are looking for blocks that have multiple child blocks. Blocks with a single child block are ignored. After encountering multiple child blocks we compare them to see whether their structure is similar. At this step we look at the structure, no content is evaluated. Structural similarity is calculated using the Sequence Matcher method [35]. We employed a slightly modified version of the algorithm with the autojunk heuristic disabled since we passed preprocessed HTML structure for analysis.



The basic schema is presented in Algorithm 3. Its principle is to compare each block with its sibling, whether they are similar or not. Experiments with different situations and their similarity estimation were conducted to measure the threshold value for similarity. The situations for experiments were selected independently from the further used web pages. We have found that a sequence matcher similarity of more than 0.92 is enough to determine whether two sibling blocks are variants of each other. Experiments with XML schema similarity [36] indicate the best results can be achieved with weight of 0.8–0.9. HTML tags are more general, therefore we increased the threshold value to 0.92. It allows interconnections and estimation of block clusters. Therefore in transformations, by applying the transformation to one of the blocks, links to other blocks exist and can be used to transform the variations of the block as well.

---

**Algorithm 3**: getVariations

---

    **input**: blocks for analysis
    **for** each block in blocks
      **call** getSiblings with block **return** siblings
      **for** each sibling in siblings
        **call** structuralSimilarity with block and sibling **return** similarity
        **if** similarity > 0.92
          **store** variation between block and sibling
        **end**
      **end**
    **end**

---

### 3.5. Novelty of the Proposed Methods

The main novelty of the paper is expressed in multiple perspectives:

- A more detailed extraction of data from content blocks is the focus of the proposed method. It not only identifies content blocks but also defines variation bounds. Such data can be used for more accurate comparisons between web page blocks.
- Methods are proposed to divide web page into content blocks. Using this approach can simplify the manual work of web page data labeling. Therefore the identified content blocks are additionally grouped to reduce the number of blocks to label. In addition, the proposed architecture allows traceability of all blocks, so labels of one element of the group can be associated with the rest of the group.
- Unique in the sense that it integrates web page block text and structure similarity. Close to Hamming distance for text similarity estimation, both parent and child relations are taken into account to identify group bounds.

In comparison to Andrew Judith et al. solution [37], our method defines as many content blocks as there are on the page, not limiting the number of blocks. In comparison to other segment number not fixed solutions [38], this method is faster, as it does not require two stages (to identify the number of clusters and then to divide the web page into this number of blocks) and extracts all possible content blocks from the web page. The blocks are not limited to text containing structured blocks only [39] and extract all, not only structured blocks [40].

## 4. Results of Web Page Division to Content Blocks

### 4.1. Data for Web Page Division to Content Blocks Validation

The validation of the proposed methods is complicated as all existing datasets are dedicated for web page segmentation and do not have extended information about block boundaries and block variations [41]. Furthermore, most of the data sources used in existing research papers are not available for repeating experiments. Nevertheless, the purpose of the methods is different, therefore, an accurate comparison would be difficult to implement. Due to the above, for the purpose of validation of the proposed solutions, a series of experiments were executed to gather the dataset.

For the experimentation, 10 existing web pages were used. Additionally, three web pages (https://1.kiril.dev/, https://5.kiril.dev/, https://6.kiril.dev/, all accessed on 27 April 2023) were prepared to reflect typical one-page websites with different content blocks. The web pages were randomly chosen from one-page website designs and stored in the selected repository to ensure they would not be modified in the future. As they were designed by web designers using the Bootstrap framework, each of them includes both the main structure of the web page and a creative approach. All the web pages were manually revised by labeling as many as possible unique content blocks.

The one-page websites or one web page of the site were chosen to illustrate a wide variety of blocks on one page. A fragment of one of the web pages and its manual block identification example is presented in Figure 5. The red border defines first-level blocks. Its internal blocks are marked in purple, while its inner blocks are presented with green borders. The example lists all content blocks and their hierarchy can be traced, while sibling block estimate (in the case of menu items, contact components or contact form fields) might reduce the need for manual segmentation actions.

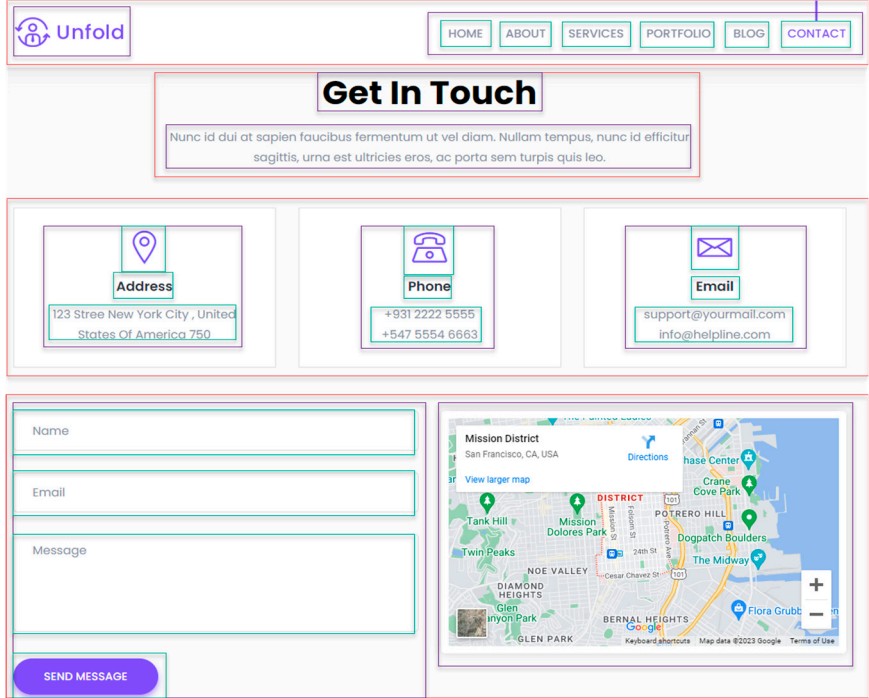

**Figure 5.** A visual view of a web page fragment with identified blocks and their hierarchy.

To label data more accurately (not only labels, but block coordinates, and block selectors are critical), a web system was created. Web page labeling participants were asked to name all content blocks they saw, including different granularity blocks. However, they were allowed to identify just one block of equivalent blocks with different content (for example, one menu item instead of all menu items one by one within the same menu). All labeled data were stored in the database for further comparison with automatically identified web page content blocks. In this study, the label of the block was not required. However, this information is stored in the same database so that it can be used in future research.

During the manual web page labeling, in total, 40,492 tags existed and 16,453 if YouTube is excluded (WYT) (see Table 2). We will further provide two values for most of the metrics, due to YouTube using a lot of proprietary tags, meaning that in some cases, statistics can be greatly affected. In any case, this amount of labeling data are too big for regular users, while expert labeling for a large number of websites might be too expensive.

**Table 2.** Summary of manually labeled data.

| No. | Web Page | No. of Tags | Labeled Blocks | Percentage of Labeled Tags |
|---|---|---|---|---|
| 1 | https://1.kiril.dev/, accessed on 27 April 2023 | 521 | 89 | 17% |
| 2 | https://5.kiril.dev/, accessed on 27 April 2023 | 531 | 84 | 16% |
| 3 | https://6.kiril.dev/, accessed on 27 April 2023 | 393 | 64 | 16% |
| 4 | https://www.youtube.com/, accessed on 27 April 2023 | 24039 | 75 | 0% |
| 5 | https://addons.mozilla.org/en-US/firefox/, accessed on 27 April 2023 | 981 | 61 | 6% |
| 6 | https://www.apple.com/, accessed on 27 April 2023 | 1087 | 69 | 6% |
| 7 | https://www.apple.com/retail/business/, accessed on 27 April 2023 | 826 | 74 | 9% |
| 8 | https://www.buzzfeednews.com/, accessed on 27 April 2023 | 1523 | 98 | 6% |
| 9 | https://gridbyexample.com/, accessed on 27 April 2023 | 111 | 17 | 15% |
| 10 | https://www.nytimes.com/, accessed on 27 April 2023 | 2474 | 203 | 8% |
| 11 | https://slack.com/, accessed on 27 April 2023 | 768 | 87 | 11% |
| 12 | https://stripe.com/en-gb-lt/connect, accessed on 27 April 2023 | 4990 | 144 | 3% |
| 13 | https://www.telegraph.co.uk/news/, accessed on 27 April 2023 | 2248 | 114 | 5% |
| | In total | 40,492 | 1179 | 3% |
| | In total without YouTube (WYT) | 16,453 | 1104 | 7% |

The labeling was conducted by persons with a basic knowledge of HTML and no experience in data labeling. They labeled 1179 (1104 WYT) blocks in total across all websites. This is just 2.9% (6.7% WYT) of the total number of HTML blocks on the surveyed web pages. Labeled data percentages across all web page tags illustrate the ratio between unique labels and tags needed to achieve a one-page website. Meanwhile, if accurate machine learning web page labeling solutions have to be created, these require a detailed dataset which would reflect all tag paths. This is an increase of labeling effort by almost 35 times or almost 15 times if we exclude YouTube data. Therefore, the labels should be duplicated or linked for different tag path variations to obtain a more accurate dataset.

### 4.2. Results of Web Page Division to Content Blocks Test Cases

The same web pages were divided into content blocks with the proposed methods. The summary of identified web page content blocks is presented in Table 3. It illustrates that the total number of content tags has been reduced by 80% (71% WYT).

**Table 3.** Summary of data for automated web page division to content blocks.

| Web Page No. | No. of Web Page Tags | No. of Potential Content Blocks | Filtered Content Blocks | Reduced Content Blocks | Boundary Range | | No. of Blocks with Siblings | Siblings | |
|---|---|---|---|---|---|---|---|---|---|
| | | | | | Single Length | Max–Min Range | | Main Sibling | Related Sibling |
| 1 | 521 | 502 | 469 | 238 | 166 | 72 | 114 | 42 | 72 |
| 2 | 531 | 518 | 494 | 192 | 88 | 104 | 91 | 24 | 67 |
| 3 | 393 | 379 | 305 | 136 | 87 | 49 | 72 | 23 | 49 |
| 4 | 24,039 | 23,983 | 5743 | 3458 | 1772 | 2816 | 621 | 226 | 395 |
| 5 | 981 | 899 | 511 | 256 | 136 | 242 | 128 | 48 | 80 |
| 6 | 1087 | 919 | 718 | 281 | 216 | 239 | 218 | 59 | 159 |
| 7 | 826 | 763 | 612 | 285 | 185 | 262 | 148 | 42 | 106 |
| 8 | 1523 | 1433 | 975 | 472 | 264 | 444 | 144 | 32 | 112 |
| 9 | 111 | 87 | 82 | 45 | 38 | 27 | 21 | 4 | 17 |
| 10 | 2474 | 2410 | 2174 | 778 | 425 | 797 | 661 | 188 | 473 |
| 11 | 768 | 702 | 364 | 168 | 116 | 137 | 111 | 29 | 82 |
| 12 | 4990 | 4375 | 2866 | 1346 | 888 | 1091 | 909 | 251 | 658 |
| 13 | 2248 | 2125 | 1344 | 436 | 184 | 557 | 223 | 55 | 168 |
| In total | 40,492 | 39,095 | 16,657 | 8091 | 4565 | 6837 | 3461 | 1023 | 2438 |
| WYT | 16,453 | 15,112 | 10,914 | 4633 | 2793 | 4021 | 2840 | 797 | 2043 |

The reduction of content blocks was achieved in several steps. We know that some tags have nothing to do with content (head tag with its contents, scripts, styles, etc.). Some of these tags (<base/>, <link/>, <meta/>, <style>) are easy to exclude, by selecting only the content of the body tag. This way, we reduce the total amount of tags from 40,492 to 39,095 (16,453 to 15,112 WYT). This amounts to a 3.5% (8% WYT) reduction. The body content should also be filtered since it usually contains tags that add value to the content. However, they do not store content themselves. For example, script tags are often included in the body tag. By filtering body content for tags that are not used to display content, we reduce the number of content tags from 39,095 to 16,657 (15,112 to 10,914 WYT). This equates to a 57% (28% WYT) reduction. In total, the reduction amounts to 59% (27% WYT) compared to the starting value of 40,492 (16,453 WYT) tags.

A more advanced reduction cannot be conducted without those simple tag reductions. Detecting content block boundaries rather than all instances of possible content tags allows us to further reduce content blocks to the mentioned 80% (71% WYT) reduction. From the filtered 16,657 (10,914 WYT) blocks, only 8091 (4633 WYT) were left by applying the proposed web page division to content blocks method. The reduction was achieved by identifying variants of different block boundaries and by leaving just one of the multiple identified sibling segments.

By grouping content block boundaries and identifying sibling variants of the block, filtered content blocks were reduced. The analysis of these two methods shows that about 51% (58% WYT) of the tags can be classified into boundary ranges. The boundary block typically groups 5 (3 WYT) tags into one block with min–max boundaries for the block.

Another form of content block reduction is the identification of relevant blocks and counting the path of one content block rather than all of them. In the analyzed web pages 3461 (2840 WYT) blocks had a sibling block. In the reduced set of content blocks, 1023 (797 WYT) were selected to represent sibling blocks, while 2438 (2043 WYT) were linked to them but eliminated. According to this, sibling blocks have, on average, three instances, but only one-third of them can be stored to represent the block pattern.

*4.3. Results of Web Page Division to Content Blocks Comparison to Manual Labeling*

The web page division to content blocks for manual and automated labeling were stored identically (except the label was not set in automated segmentation) in the same database structure but in different instances of it. Due to the database structure matching, a comparison of manual and automated division is possible. On the other hand, it is not a straightforward process, as in manual labeling, the user could identify some tags and content blocks but not others. The grouping of tags into maximum and minimum boundary blocks was not requested either. Therefore, additional methods were prepared to match a manually labeled tag to an automated division to content block with an estimation of whether the manually labeled tag fits within the content block boundaries (minimum bounds <= labeled tag <= maximum bounds). By using this method, we can estimate the match between labels, even in cases where the boundaries are labeled similarly but not identically. Another method estimates whether a labeled tag corresponds to another content block based on its structure. The feature gathers data about sibling blocks, which were ignored for simplicity, however, correspond to some of the already identified blocks but with different content.

This matching between automated web page division to content blocks and manual labeling enabled us to estimate whether our solution was able to identify all content blocks labeled by humans. In total, 1179 content blocks were labeled by human experts (1104 WYT). The dependency of a number of tags in the web page and our method of identified blocks are presented in Figure 6 which indicates the linear dependency.

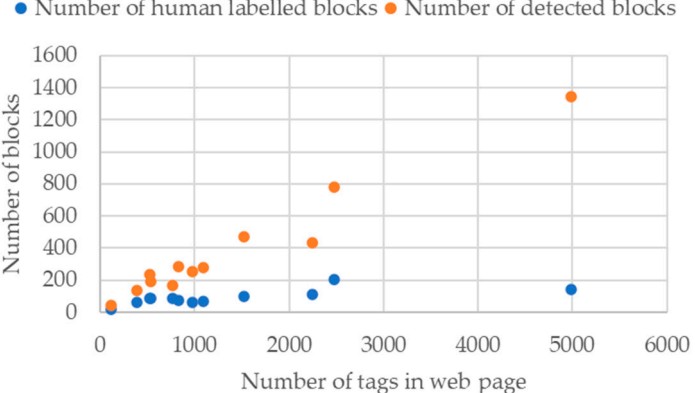

**Figure 6.** Dependency between human-labeled number of blocks and tags in the appropriate web page.

The labeled blocks were mapped to the identified by using our proposed content block identification method. For further performance analysis, the standard classification metrics were used. The true positive (TP) was assumed for the number of blocks, indicated by our method and matching the dataset-defined segments or user-labeled blocks. False positive (FP) were the other blocks our method detected, but which were not labeled in the dataset. False negative (FN) was for a number of blocks which were labeled in the dataset but were missing in our output, while the true negative (TN) was calculated by subtracting TP, FP, and FN from the total number of tags in the web page. The summary of the metrics is presented in Table 4.

**Table 4.** Summary of proposed method accuracy metrics comparing reduced content blocks.

| Web Page No. | Number of Blocks | Number of | | | | Precision | Recall | Accuracy | F-Score |
|---|---|---|---|---|---|---|---|---|---|
| | | True Positive | True Negative | False Positive | False Negative | | | | |
| 1 | 238 | 89 | 0 | 149 | 0 | 37% | 100% | 71% | 54% |
| 2 | 192 | 84 | 0 | 108 | 0 | 44% | 100% | 80% | 61% |
| 3 | 136 | 64 | 0 | 72 | 0 | 47% | 100% | 82% | 64% |
| 4 | 3458 | 75 | 0 | 3383 | 0 | 2% | 100% | 86% | 4% |
| 5 | 256 | 61 | 0 | 195 | 0 | 24% | 100% | 80% | 38% |
| 6 | 281 | 69 | 0 | 212 | 0 | 25% | 100% | 80% | 39% |
| 7 | 285 | 74 | 0 | 211 | 0 | 26% | 100% | 74% | 41% |
| 8 | 472 | 98 | 0 | 374 | 0 | 21% | 100% | 75% | 34% |
| 9 | 45 | 17 | 0 | 28 | 0 | 38% | 100% | 75% | 55% |
| 10 | 778 | 203 | 0 | 575 | 0 | 26% | 100% | 77% | 41% |
| 11 | 168 | 87 | 0 | 81 | 0 | 52% | 100% | 89% | 68% |
| 12 | 1346 | 144 | 0 | 1202 | 0 | 11% | 100% | 76% | 19% |
| 13 | 436 | 114 | 0 | 322 | 0 | 26% | 100% | 86% | 41% |
| Overall | 8091 | 1179 | 0 | 6912 | 0 | 15% | 100% | 83% | 25% |
| WYT | 4633 | 1104 | 0 | 3529 | 0 | 24% | 100% | 79% | 38% |

The results indicate that the proposed web page division to content blocks solution can identify all content blocks that would be manually labeled. At the same time, it identifies additional content blocks that were ignored during manual labeling. The main reason for ignoring some blocks during manual labeling is their repetitive nature. This repetitiveness can be observed in a couple of ways. First, when a block can have various boundaries while presenting the same content, we use the block's max–min boundary detection to negate the need to label all of the possible combinations of the same block. Another situation with repetitive blocks arises when there are multiple content blocks for the same purpose but with different content. The most basic example of this is the navigation menu. Each menu element has the same structure as all the other elements of the same menu, so users tend to mark only one menu element. We use structural sibling relationships between blocks to detect other menu elements. This allows us to detect all menu items regardless of which menu element was labeled by the user. Sometimes, these two cases of repetitiveness can

happen at the same time; for example, menu items can also have multiple boundaries within the max–min range, so both techniques can be used at the same time to determine all other possible labeled block variations.

The obtained results of automated web page division to content blocks comparison to manual labeling results indicate that the solution can identify all manually labeled blocks (directly or indirectly, by using related sibling records). This leads to 100% precision. Currently, the increase in testing data is problematic. This is because the existing datasets of web page segmentation or labeling are not fully adapted to the extended model of web page labeling.

Talking about the accuracy of the proposed method, it could be expressed as 83% (79% WYT) taking into account how many blocks were labeled by person and were present in the dataset of automatically detected blocks, eliminating relevant siblings. These conditions correspond to the ones that were presented for manual labeling—labeling all components for the same purpose. Under the same conditions, the F-score would be 25% (38% WYT).

### 4.4. Proposed Method Comparison with Existing Segmentation Methods

To compare the proposed method with other existing web page division to content blocks is complicated as there are no exact analogues. However, web page segmentation solutions are very similar by their nature. Those methods are mostly validated by using a commonly used dataset [23]. The labeling of the dataset is not as broad as the proposed method aims to provide. It reflects both in the number of blocks (the average number of blocks in the dataset is 13, while our previously tested web pages had an average of 79 blocks) and details about each block (the dataset has specific block boundaries, while our solutions and previously used web page analysis data has minimum and maximum ranges for each block, as well as relations between siblings, similar blocks). While this dataset has a much higher number of records, web pages usually estimate the method's performance by using this dataset.

This dataset was selected as some existing web page segmentation methods already used it, therefore, there are accuracy metrics for those methods (see Table 1 in Section 2). The precision, recall, accuracy, and F-score were calculated for each record in the dataset, and average values were calculated to summarize the results. With this dataset, our proposed method achieved 11% precision, 100% recall, and 77% accuracy, and the F-score was 19%. The results indicate that our proposed method is not precise (11%), and the F-score (19%) is the lowest among other methods. However, it is related to the fact that the dataset contains just a small portion of labeled blocks and segments, while our solutions aim to find all possible content blocks. Moreover, the numbers are not directly comparable as the other research papers were estimating the accuracy of used segments, not content blocks. However, even taking into account our method of grouping sibling elements into groups, it shows a high accuracy (77%) similar to the web page segmentation methods, while the recall stays constant (100%) because no blocks are removed from the web page, just assigned to one or another block variation or group.

## 5. Discussion

In the proposed solution, DOM tree and web page similarity estimation are used instead of a visual comparison of the web page. This simplifies its application as no complex models for data clustering are needed.

While our web page division to content blocks method results cannot be compared directly with other research results (because of different purpose, data, and dataset labeling details), they are similar to those obtained by other web page labeling or segmentation methods [12]. As the results can differ depending on the dataset and labeling, the proposed method was compared to a dataset [23] and methods used to segment the same dataset. According to the obtained results, we obtain lower results, but that is due in part to the fact that the dataset had a very limited number of labels, and is adapted to present segments, not all content blocks.

The precision value is among the lowest in comparison to existing research and experiment results. This is affected by the limited number of labeled blocks in the dataset as well, as only the main blocks are included in the dataset but not all small elements were labeled internally. Recall is the other side of it. Our methods achieve 100% recall and outperform any other method. This is because we do not eliminate any of the blocks but group them into variations or sibling groups.

Summarizing the results and limitations of the method, its high potential could be exploited for assistance in manual web page labeling. The method could extract all possible content blocks from the web page and present them to the individual executing the labeling. This would reduce the need for tag/block revision by 90% (70% not taking into account the YouTube case). At the same time, the labeling data will be richer in the sense of relations between blocks. This could be exploited even for more interactive labeling when label assignment to one block automatically generates proposals for the labels of other related blocks.

As more detailed datasets of the extended labeling data become available, the method could be improved to identify or propose the label for the block. This would lead to a full understanding of the web page structure, therefore, the automated integration and transformation of web page content would be possible.

**Author Contributions:** Conceptualization, K.G. and S.R.; methodology, S.R.; software, K.G.; validation, K.G.; formal analysis, S.R.; investigation, K.G.; resources, K.G.; data curation, K.G.; writing—original draft preparation, K.G.; writing—review and editing, S.R.; visualization, K.G.; supervision, S.R. All authors have read and agreed to the published version of the manuscript.

**Funding:** This research received no external funding.

**Institutional Review Board Statement:** Not applicable.

**Informed Consent Statement:** Not applicable.

**Data Availability Statement:** The data can be requested from the corresponding author.

**Conflicts of Interest:** The authors declare no conflict of interest.

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
