# Peer review of "Web Page Content Block Identification with Extended Block Properties"

_applsci, doi:10.3390/app13095680_

Round 1
Reviewer 1 Report
The paper proposes a simple method of web page segmentation based on the analysis of HTML element nesting and content similarity. Although there are nice aspects in the method such as the detection of "variations", the overall approach seems to be very simple resembling the first DOM-based methods introduced many years ago. In my opinion, the paper has the following main problems:
- It is not explicitly defined what the authors mean by page segmentation, what properties the detected segments should have, and how they should be organized. At least it seems that the authors have a different understanding of page segmentation than the rest of the referenced methods (i.e., finding the page blocks that are semantically or at least visually consistent), because consistency and other aspects are not evaluated.
- Since the requirements for the detected blocks are not defined, it is not possible to assess whether the results obtained meet the objectives. The authors do not mention any advantages/disadvantages of the existing methods that should be addressed, and therefore it is not clear in which aspect the proposed method is considered to perform better.
- Using only three pages for evaluation is completely inadequate. The HTML code can be very variable, and the method is based on many assumptions about code structure, use of empty elements, and other aspects that should be properly verified. For example, it is not clear what happens when advanced layout methods such as flexbox or grid (or even absolute positioning) are used.
- The authors cite the work of Kiesel et al. which provides a large dataset of manually segmented pages (ground truth) as well as the results of several segmentation algorithms, and they also define the way to compare the results. However, for some reason, this newer dataset was not used for the evaluation. Instead, the authors focus on comparing the number of segments detected, which does not seem to be the most important feature of a segmentation algorithm. The authors also mention the precision and recall of the method, but the meaning of these metrics is not defined in the given context (what is considered a true positive, false positive, etc. in the context of page segmentation?) I wonder why the existing results (since the evaluation tools seem to be publicly available) were not used to compare the new method with the existing ones in the same way.
In conclusion, although some ideas are interesting, in my opinion the proposed method tends to ignore the research in this area over the last 15 years, it is not compared to existing methods, nor is it clear what benefits it is expected to bring. Therefore, I do not find the paper suitable for journal publication.
Author Response
Thank you for your comments. We appreciate your opinion, but at the same time would like to stress out some points, which illustrate the benefit and novelty of the research, as well as address what changes were made.
The comments and our responses are presented in the attached file.
As well the changes in the paper are highlighted with red font.

Reviewer 2 Report
Web Page Content Block Segmentation with Extended Block 2
Properties
comment -1
Abstract needs to be rewritten to m
DOM has to be extended
commnet -2
Introduction has to be revised. The statements given the introduction
have to be justified from literature references.
For example, the contents of the lines in 25 and 26
(Machine-adapted labelling of content on web pages is a 25
base for automated content extraction)
should be supported with literature reference
contents of lines 34-35 (
The first part is done by segmenting the HTML code, while the second part uses intelligent 34
solutions to classify the blocks into predefined types. ) -- provide literature support
comment -3
line 35-36 ( However both of these parts currently are facing some challenges)
authors are asked to mention the challenges
also, include literature support for the above challenges
comment - 4
some blocks might be divided into smaller internal blocks and the level of detail- 37
ing might cause redundancy of blocks; content block might have HTML code variations, 38
caused by its presentation structure, therefore for more accurate block classification the 39
possible ranges would be beneficial; hierarchical and sibling relations between different 40
blocks might positively affect the block classification accuracy and lead to a reduced size 41
of the dataset not losing the data itself.
The intended meaning of the above contents is not clear. can be rewritten
comment - 5
line 59 - expand VIPS
comment -6
line 65 and 66 are not related and can be given through separate sentences
comment -7
Existing methods do not focus on the range of HTML 73
code corresponding to the same segment, and segment variations are not linked to getting 74
a more interconnected segment map.
the authors are suggested to describe what exactly 'range' means?
comment -8
section -3. Figure 1 caption needs to be modified
comment -9
the authors have described the meaning of range with a very simple block
but in reality, in a html file, there may be many child tags withing
maximum boundaries. In addition, a tag (child tag) of a parent
may contain other child tags. i.e. tags can be nested.
In the above case, how to define range. The authors are suggested
provide a more generic example
comment - 10
We have found that a sequence matcher similarity of more than 0.92 is enough to deter- 248
mine whether two sibling blocks are variants of each other. I
How 0.92, the authors are suggested to provide evidence from literature
comment -11
Methods are proposed to segment web page blocks to simplify labelling 261
manual web page data. (line 261-262) needs to be rewritten
Author Response
Thank you for your comments. We appreciate your opinion and proposals. We addressed the issues and updated the paper.
The comments and our responses are presented in the attached file.
As well the changes in the paper are highlighted with red font.

Reviewer 3 Report
Add a novelty section that highlights the significance of your study
Author Response
Thank you for your comment.
Section 3.5 summarizes the novelty of our proposed methods. As well additional changes in the paper were done and are highlighted with red font.
Round 2
Reviewer 1 Report
I confirm having read the authors' response to the previous review and I think the purpose of the proposed method is now better explained and seems reasonable. On the other hand, in the related work and the experimental evaluation, the proposed method is still compared with page segmentation methods even though the purpose is different. At the very least, the motivation and difference in requirements, which was explained in the authors' response should be explained in the paper as well and demonstrated by the experimental results.
I still insist that it is not sufficient to use 3 pages for evaluation, especially if they have been tailored for this single purpose. The method should be verified on real web pages and it is not enough to write that it would probably perform similarly even if grid, flexbox or absolute positioning were used. This has to be proven by the experiments.
Author Response
10 additional web sites were analyzed.
The related works were extended as well.
Reviewer 2 Report
Thoroughly check for the clarity of presentation
Author Response
Thank you for the comment. Additional visualization was added.
Reviewer 3 Report
NA
Author Response
Thank you for your time, reviewing the paper.